# A Molecular Electron Density Theory Study of the Synthesis of Spirobipyrazolines through the Domino Reaction of Nitrilimines with Allenoates

**DOI:** 10.3390/molecules24224159

**Published:** 2019-11-16

**Authors:** Luis R. Domingo, Fatemeh Ghodsi, Mar Ríos-Gutiérrez

**Affiliations:** 1Department of Organic Chemistry, University of Valencia, Dr Moliner 50, 46100 Burjassot, Valencia, Spain; rios@utopia.uv.es; 2Department of Chemistry, University of Sistan and Baluchestan, Zahedan 98135-674, Iran; fa.ghodsi@yahoo.com; 3Department of Chemistry and Chemical Biology, McMaster University, 1280 Main Street West, Hamilton, ON L8S 4L8, Canada

**Keywords:** [3+2] cycloaddition reactions, nitrilimines, allenoates, spirobipyrazolines, domino reactions, molecular electron density theory, molecular mechanism

## Abstract

The reaction of diphenyl nitrilimine (NI) with methyl 1-methyl-allenoate yielding a spirobipyrazoline has been studied within molecular electron density theory (MEDT) at the MPWB1K/6-311G(d) computational level in dichloromethane. This reaction is a domino process that comprises two consecutive 32CA reactions with the formation of a pyrazoline intermediate. Analysis of the relative Gibbs free energies indicates that both 32CA reactions are highly regioselective, the first one being also completely chemoselective, in agreement with the experimental outcomes. The geometries of the TSs indicate that they are associated to asynchronous bond formation processes in which the shorter distance involves the C1 carbon of diphenyl NI. Despite the zwitterionic structure of diphenyl NI, the appearance of a *pseudoradical* structure at the beginning of the reaction path, with a very low energy cost, suggests that the 32CA reaction between diphenyl NI, a strong nucleophile, and the allenoate, a moderate electrophile, should be mechanistically considered on the borderline between *pmr-type* and *cb-type* 32CA reactions, somewhat closer to the latter.

## 1. Introduction

Spirocyclic compounds are fused bicyclic structures sharing a central carbon (see [Fig molecules-24-04159-ch001]). They have received considerable attention because of their abundant presence in natural products and their interesting conformational features [1,2,3]. Spiroheterocyclic compounds have been found to exhibit valuable biological activities as well as pharmacological and therapeutical properties (see [Fig molecules-24-04159-ch001]) [4,5,6,7,8,9].

One of the most common methods for the construction of the spirocyclic framework involves the formation of a new ring on an existing carbo- or heterocycle. Few examples of formation of spirocyclic compounds through a double-intramolecular [3+2] cycloaddition (32CA) reaction of an allene in a one-pot reaction have been reported [10,11,12]. Although several cycloaddition reactions of allenoates with a three-atom-component (TAC) have been accomplished, only one of the two C–C double bonds was involved in these reactions [13,14,15,16,17,18].

Given the interesting biological activities of spiropyrazolines, their synthesis has attracted great attention. Several studies for the synthesis of spiropyrazolines have been reported [19]. In general, the synthesis of these compounds involves 32CA reactions [20,21,22,23,24,25,26,27,28,29,30,31,32,33,34] or condensation reactions [35,36,37] as the essential step. Several examples of TACs used in the synthesis of spiropyrazolines via 32CA reactions, including nitrilimines [20,21,22,23,24], diazoalkanes [25,26,27], related diazo derivatives [28,29,30,31] and diaziridines [32,33,34], have been reported.

Nitrilimines (NIs) are important in situ generated TACs widely used in the synthesis of pyrazoline structures [38]. The use of heterocyclic *α*,*β*-enone compounds presenting exocyclic double bonds has allowed the synthesis of spiropyrazolines [38]. In 2014, Santos et al. reported the synthesis of spiropyrazoline **3** via the 32CA reaction of diphenyl NI **1** with 3-methylene indolinone **2** having an exocyclic double bond (Scheme 1) [39]. This reaction was completely regioselective. Formation of spiropyrazoline **3** involved the formation of the C–C single bond between the carbon center of diphenyl NI **1** and the *β*-conjugated carbon of the exocyclic C–C double bond of 3-methylene indolinone **2**.

Recently, Gou et al. reported, for the first time, the synthesis of spirobipyrazoline **5** via a double 32CA reaction of diphenyl NI **1** with methyl 1-methyl-allenoate **4** (Scheme 2) [40].

Formation of spirobipyrazoline **5** is a domino process that comprises two consecutive 32CA reactions of two molecules of diphenyl NI **1** with the two C–C double bonds of allenoate **4** (see Scheme 3). Formation of a single spirocompound **5** indicates that the two 32CA reactions are completely chemo- and regioselective. Gou et al. proposed a plausible mechanism for this domino process in which diphenyl NI **1**, generated in situ, reacts chemoselectively with the *α*,*β*-conjugated C–C double bond of allenoate **4** to yield pyrazoline **6**, which was observed on the ^1^H NMR spectra of the reaction [40]. Then, this compound experiences a second 32CA reaction between the exocyclic C–C double bond of pyrazoline **6** and a second molecule of diphenyl NI **1**, yielding the final spirobipyrazoline compound **5**.

The 32CA reactions of the simplest NI **7** with ethylene **8** and electrophilic dicyanoethylene (DCE) **10** have recently been studied within the molecular electron density theory [41] (MEDT) (see Scheme 4) [42]. Topological analysis of the electron density of the simplest NI **7** showed that this TAC has a carbenoid structure participating in a *cb-type* 32CA reaction. The activation energy of the 32CA reaction of the simplest NI **7** with DCE **10** was 5.6 kcal·mol^−1^ lower than that involving ethylene **8**, 10.2 kcal·mol^−1^, in agreement with the high polar character of the 32CA reaction [42]. The 32CA reaction with DCE **10** presented a low 1,5-regioselectivity, yielding pyrazoline **12** as the major product.

In view of the interesting carbenoid structure of the simplest NI **7** participating in *cb-type* 32CA reactions, and the complete regioselectivity found in the two consecutive 32CA reactions involved in the formation of spirobipyrazoline **5**, see Scheme 3, an MEDT study of the domino reaction of diphenyl NI **1** with allenoate **4**, experimentally reported by Gou et al. [40] is herein carried out. The main purpose of this MEDT study is to shed light on the reactivity of diphenyl NI **1** towards the two C–C double bonds of allenoate **4**, which determine the chemo- and regioselectivity in the first 32CA reaction, and the regioselectivity in the second 32CA reaction.

## 2. Computational Methods

DFT calculations were performed using the MPWB1K functional [43] together with the 6-311G(d) basis set [44]. Optimisations were carried out using the Berny analytical gradient optimisation method [45,46]. The stationary points were characterized by frequency computations in order to verify that TSs have one, and only one, imaginary frequency. The intrinsic reaction coordinate [47] (IRC) paths were traced in order to check the energy profiles connecting each TS to the two associated minima of the proposed mechanism using the second order González-Schlegel integration method [48,49]. Solvent effects of dichloromethane (DCM) were taken into account by full optimization of the gas phase structures using the polarisable continuum model (PCM) [50,51] in the framework of the self-consistent reaction field (SCRF) [52,53,54]. Values of enthalpies, entropies and Gibbs free energies were calculated with standard statistical thermodynamics [44] at reaction conditions: 25 °C and 1 atm in DCM [40], using the already optimized geometries including solvent effects.

The global electron density transfer [55] (GEDT) was computed by the sum of the natural atomic charges (q), obtained by a Natural Population Analysis [56,57] (NPA), of the atoms belonging to each framework (f) at the TSs; i.e., GEDT (f) = $\underset{q\in f}{\sum }q.$ The sign indicates the direction of the electron density flux in such a manner that positive values mean a flux from the considered framework to the other one. Conceptual DFT (CDFT) global reactivity indices [58,59] and Parr functions [60] were computed using the equations given in reference 59. All computations were carried out with the Gaussian 16 suite of programs [61].

Topological analysis of the Electron Localisation Function [62] (ELF) was performed with the TopMod [63] package using the corresponding gas phase monodeterminantal wavefunctions and considering the standard cubical grid of step size of 0.1 Bohr. The characterization of the bond formation processes along the most favourable reaction path was carried out by applying the bonding evolution theory (BET) [64] to the IRC profile. The molecular geometries and ELF basin attractor positions were visualised using the GaussView program [65].

## 3. Results and Discussion

The present MEDT study has been divided in four sections: (i) in Section 3.1, the electronic structure of diphenyl NI **1** and allenoate **4** is analysed by means of the topological analysis of the ELF; (ii) in Section 3.2, an analysis of the CDFT reactivity indices of the reagents is carried out; (iii) in Section 3.3, the reaction paths associated with the domino reaction between diphenyl NI **1** and allenoate **4** are investigated; and finally, (iv) in Section 3.4, the formation of the C–C and C–N single bonds along the most favourable reaction path associated with the 32CA reaction of diphenyl NI **1** with allenoate **4** is characterized.

### 3.1. Topological Analysis of the ELF of Diphenyl NI **1** and Methyl 1-Methyl-Allenoate **4**

An appealing procedure that provides a straightforward connection between the electron density distribution and the chemical structure is the quantum chemical analysis of Becke and Edgecombe’s ELF [62]. As commented above, the simplest NI **7** has a carbenoide structure, which is topologically characterised by the presence of one V(C) monosynaptic basin integrating 1.66 e [41]. Note that monosynaptic basins are associated to a single nucleus and, therefore, to non-bonding electron density, while the basin population of ca. 2 e would be related to a lone pair in the Lewis boding model. This structural behaviour makes the participation of the simplest NI **7** in *cb-type* 32CA reactions possible [66]. Herein, in order to characterize the electronic structure of diphenyl NI **1** and, thus, to establish its reactivity in 32CA reactions, a topological analysis of the ELF of this TAC, as well as of allenoate **4**, was first performed. ELF basin attractor positions together with the valence basin populations, as well as the proposed Lewis-like structures together with the natural atomic charges, of diphenyl NI **1** and allenoate **4** are shown in Figure 1.

Topological analysis of the ELF of diphenyl NI **1** shows the presence of one V(N3) monosynaptic basin integrating 3.37 e, one V(N2,N3) disynaptic basin integrating 2.40 e, and a pair of disynaptic basins, V(C1,N2) and V’(C1,N2), integrating 2.44 and 2.77 e, respectively. No V(C1) monosynaptic basin characterizing a carbenoid structure is observed in diphenyl NI **1**. The population of the V(C1,CAr) disynaptic basin, which integrates 2.93 e, together with that associated with the C1–N2 bonding region, 5.21 e, indicate that the C1 carbenoid center present in the simplest NI **7** has been delocalised in the N2–C1–CAr bonding region of diphenyl NI **1**. This behaviour is similar to that observed in substituted *pseudodiradical* and *pseudoradical* TACs in which the presence of a phenyl substituent at the *pseudoradical* carbon center delocalises the corresponding non-bonding electron density into the phenyl ring [67].

Interestingly, topological analysis of the ELF of dimethyl NI **13** shows the presence of a V(C1) monosynaptic basin integrating 1.51e, characterising the C3 carbon as a carbenoid center, while the C1–N2 and C1–C(Me) bonding regions have been depopulated by 0.63 and 0.96 e, respectively, with respect to the equivalent regions of diphenyl NI **1**. These changes in the electronic structure of these NIs with the phenyl substitution are reinforced with the analysis of the geometries of these NIs: while diphenyl NI **1** presents a linear propargylic structure with an N2–C1–C(Ar) bond angle of 178.2 degrees, dimethyl NI **13** presents a linear allenic structure with an N2–C1–C(Me) bond angle of 141.5 degrees and a C(Me) –N3–C1–C(Me) dihedral angel of −96.6 degrees [42].

On the other hand, methyl 1-methyl-allenoate **4** is topologically characterised by the presence of two pairs of disynaptic basins, V(C4,C5) and V’(C4,C5), and V(C5,C6) V’(C5,C6), both integrating a total population of 3.70 e, which are therefore related to two C4–C5 and C5–C6 double bonds, and a V(C7,O8) disynaptic basin, integrating 2.45 e, associated with the polarised carbonyl C–O bonding region of allenoate **4**, which should rather be considered a single bond.

NPA of NIs **1** and **13** reveals a similar electron density distribution. In diphenyl NI **1**, while the N3 nitrogen is negatively charged by 0.37 e, the N2 nitrogen presents negligible charges of 0.02 e and the C3 carbon is positively charged by 0.32 e (see Figure 1). Note that the central N2 nitrogen has a negligible charge in both NIs. The most relevant difference is that the C1 carbon is more positively charged at NI **1** than at NI **13**, in agreement with the delocalising effect exerted by the phenyl substituent. This charge distribution, resulting from the molecular electron density distribution, does not account for Huisgen’s zwitterionic representation of the electronic structure of these TACs [68].

### 3.2. Analysis of the CDFT Reactivity Indices of the Reagents

Numerous studies devoted to Diels-Alder and 32CA reactions have shown that the analysis of the reactivity indices defined within CDFT [58,59] is a powerful tool to predict and understand the reactivity in polar and ionic organic reactions. Thus, in order to predict the reactivity of diphenyl NI **1** in the 32CA reaction with allenoate **4**, the global CDFT reactivity indices, i.e., the electronic chemical potential, μ, the chemical hardness, η, the electrophilicity, ω, and the nucleophilicity, *N*, at the ground state of the reagents, were computed and analysed (see Table 1).

The electronic chemical potential μ [69] of the diphenyl NI **1**, −3.38 eV, is higher than that of allenoate **4**, −4.18 eV, indicating that along a polar 32CA reaction the electron density will flux from diphenyl NI **1**, thus acting as the nucleophile, towards allenoate **4**, acting as the electrophile.

The electrophilicity [70] and nucleophilicity *N* [71] indices of diphenyl NI **1** are 1.13 and 4.89 eV, respectively, being classified as a moderate electrophile and a strong nucleophile within the electrophilicity and nucleophilicity scales [59]. Substitution of the two phenyl groups in NI **1** by two methyl groups in dimethyl NI **13** decreases the electrophilicity ω and nucleophilicity *N* indices to 0.57 and 4.22 eV, respectively. Now, while NI **13** keeps being a strong nucleophile, it becomes a marginal electrophile.

On the other hand, allenoate **4** has an electrophilicity ω index of 1.05 eV and a nucleophilicity *N* index of 2.45 eV, being classified as a moderate electrophile and a moderate nucleophile.

Despite the similar electrophilic character of diphenyl NI **1** and allenoate **4**, the strong nucleophilic character of the former indicates that along a polar interaction diphenyl NI **1** will act as nucleophile and allenoate **4** as electrophile, in agreement with the analysis of the electronic chemical potential μ of both reagents. However, it is expected that the corresponding 32CA reaction will have a low polar character because of the moderate electrophilic character of allenoate **4**.

By approaching a non-symmetric electrophilic/nucleophilic pair along a polar or ionic process, the most favourable reactive channel is that associated with the initial two-center interaction between the most electrophilic center of the electrophile and the most nucleophilic center of the nucleophile. In this context, the nucleophilic $P_{k}^{-}$ and electrophilic $P_{k}^{+}$ Parr functions have shown to be the most accurate and insightful tools for the study of the local reactivity in polar and ionic processes [60]. Therefore, the nucleophilic $P_{k}^{-}$ Parr functions of diphenyl NI **1** and dimethyl NI **13**, and the electrophilic $P_{k}^{+}$ Parr functions of allenoate **4** were analysed in order to characterise the most nucleophilic and electrophilic centers of these species (see Figure 2).

Analysis of the nucleophilic $P_{k}^{-}$ Parr functions at the reactive sites of diphenyl NI **1** indicates that the terminal N3 nitrogen is the most nucleophilic center of this molecule $P_{k}^{-}$(N3) = 0.47. Note that it is twice as nucleophilically activated as the C1 carbon, $P_{k}^{-}$(C1) = 0.21. Interestingly, analysis of the nucleophilic $P_{k}^{-}$ Parr functions at the reactive sites of dimethyl NI **13** indicates that both the C1 carbon, $P_{k}^{-}$(C1) = 0.57, and the terminal N3 nitrogen $P_{k}^{-}$(N3) = 0.63, present similar nucleophilic activation (see Figure 2), just as in the case of the simplest NI **7** [42].

Conversely, the electrophilic $P_{k}^{-}$ Parr functions at allenoate **4** show that the central C5 carbon of the allenic system is the most electrophilic center of this molecule, $P_{k}^{-}$ = 0.51, while the other terminal C4 and C6 allene carbon atoms are scarcely activated, 0.07, or even electrophilically deactivated, −0.13. Interestingly, the carbonyl C7 carbon is half as electrophilically activated as the allene C5 carbon, $P_{k}^{-}$ = 0.24, which is still significant. However, note that the carbonyl C7 carbon does not participate in any of the two 32CA reactions.

Consequently, it is expected that along a polar reaction, the most favourable reaction path will correspond to the two-center interaction between the N3 nitrogen of diphenyl NI **1** and the C5 carbon of allenoate **4**, against the experimental outcomes.

### 3.3. Study of the Reaction Paths Associated with the Domino Reaction of Diphenyl NI **1** with Allenoate **4**

The reaction of diphenyl NI **1** with allenoate **4** yielding spirobipyrazoline **5** is a domino process that comprises two consecutive 32CA reactions of two molecules of NI **1** with the two C5–C4(6) double bonds of the allenic system of allenoate **4** (see Scheme 5). Due to the non-symmetry of the two reagents, and the presence of two C–C double bonds in allenoate **4**, four competitive reaction paths are feasible for the first 32CA reaction of this domino process; being related to two chemo- and two regioisomeric reaction paths. For the second 32CA reaction, only two regioisomeric reaction paths are feasible for each of the pyrazolines formed along the first 32CA reaction (see Scheme 5). Relative enthalpies and Gibbs free energies in DCM are given in Scheme 5, while the absolute enthalpies and Gibbs free energies in DCM are given in Appendix A in ESI.

For the first 32CA reaction of diphenyl NI **1** with allenoate **4**, the activation enthalpies associated with the four competitive reaction paths are: 10.6 (**TS11**), 14.0 (**TS11r**), 17.0 (**TS21**) and 17.6 (**TS21r**) kcal·mol^−1^; formation of the corresponding pyrazolines is strongly exothermic between 69.8 (**6**) and 80.2 (**14**) kcal·mol^−1^. Some appealing conclusions can be drawn from these energy results: (i) the most favourable reaction path, which is associated with the formation of pyrazoline **6** via **TS11**, presents an activation enthalpy of 10.6 kcal·mol^−1^. This value is 2.7 kcal·mol^−1^ higher than that associated with the 32CA reaction of carbenoid dimethyl NI **13** with allenoate **4** (see Scheme S1 in ESI). This reaction path is associated with the attack of the C1 carbon of diphenyl NI **1** on the *β*-conjugated C5 carbon of allenoate **4**; (ii) the attack of NI **1** on the *α*,*β*-conjugated C4–C5 double bond of allenoate **4** is completely regioselective as **TS11r** is 3.4 kcal·mol^−1^ above **TS11**. Note that the analysis of the nucleophilic $P_{k}^{-}$ Parr functions at the reactive sites of diphenyl NI **1** predicts this reaction path to be less favourable than its regioisomeric one; (iii) this 32CA reaction is completely chemoselective as **TS21** is 6.4 kcal·mol^−1^ above **TS11**; (iv) formation of pyrazoline **6** is strongly exothermic by 69.8 kcal·mol^−1^; and (v) these energy results rule out the formation of pyrazoline **14** along this domino reaction, in complete agreement with the experimental observation of pyrazoline **6** as the only transient species in the reaction media [40].

For the second 32CA reaction, two regioisomeric reaction paths are feasible for each pyrazoline. However, only those associated to the attack of diphenyl NI **1** on pyrazolines **6** and **14**, which enable the formation of the final spirobipyrazoline **5**, were considered (see Scheme 5). The activation enthalpies associated with the two pairs of competitive reaction paths of these 32CA reactions are: 7.9 (**TS12**), 10.5 (**TS12r**), 13.7 (**TS22**) and 12.4 (**TS22r**) kcal·mol^−1^; formation of the corresponding spiro compounds is strongly exothermic by between 55.0 (**17**) and 57.2 (**5**) kcal·mol^−1^. Some appealing conclusions can be drawn from these energy results: (i) as expected, the most favourable 32CA reaction is that associated with the nucleophilic attack of the N1 nitrogen of diphenyl NI **1** on the *β*-conjugated position of pyrazoline **14**. However, this reaction path is not feasible because **14** is not formed along the first 32CA reaction; (ii) the 32CA reaction of diphenyl NI **1** with pyrazoline **6** is highly regioselective as **TS12r** is 2.7 kcal·mol^−1^ higher in energy than **TS12**; (iii) the activation enthalpy associated with the attack of diphenyl NI **1** on the *α*,*β*-conjugated C4–C5 double bond of pyrazoline **14**, 13.7 kcal·mol^−1^ (**TS22**), is 3.1 kcal·mol^−1^ higher in energy than the attack on the *α*,*β*-conjugated C4–C5 double bond of allenoate **4** via **TS11**. On the other hand, the activation enthalpy associated with the attack of diphenyl NI **1** on the exocyclic C5–C6 double bond of pyrazoline **6**, 7.8 kcal·mol^−1^ (**TS12**), is 9.2 kcal·mol^−1^ lower in energy than the attack on the C5–C6 double bond of allenoate **4** via **TS21**; and finally, (iv) from pyrazoline **6**, formation of spirobipyrazoline **5** is exothermic by 57.2 kcal·mol^−1^, 12.6 kcal·mol^−1^ less exothermic than the first 32CA reaction involving allenoate **4**.

The Gibbs free energy profiles associated to the competitive reaction paths of the domino reaction between diphenyl NI **1** and allenoate **4** yielding spirobipyrazoline **5** are given in Figure 3. The activation Gibbs free energy associated to the more favourable regioisomeric reaction path for the attack of diphenyl NI **1** on the *α*,*β*-conjugated C4–C5 double bond of allenoate **4**, via **TS11**, is 27.5 kcal·mol^−1^, formation of pyrazoline **6** being strongly exergonic by 51.7 kcal·mol^−1^. From pyrazoline **6**, formation of spirobipyrazoline **5**, via **TS12**, presents an activation Gibbs free energy of 24.5 kcal·mol^−1^, being exergonic by −61.4 kcal·mol^−1^. Formation of spirobipyrazoline **5** from diphenyl NI **1** and allenoate **4** is strongly exergonic by 90.4 kcal·mol^−1^.

Some appealing conclusions can be drawn from these Gibbs free energy profiles: (i) the initial attack of diphenyl NI **1** on allenoate **4** is highly regioselective and completely chemoselective as **TS11r** and **TS21** are 3.0 and 4.2 kcal·mol^−1^ higher in Gibbs free energies than **TS11**, respectively. Note that considering the aforementioned ΔΔG^≠^ between the regioisomeric **TS11** and **TS11r** and the reaction conditions, 25 °C in DCM, a 99.3:0.7 ratio of pyrazolines **6** and **14** is estimated using the Eyring-Polanyi equation [72]; (ii) the high exergonic character of the formation of pyrazoline **6**, −51.7 kcal·mol^−1^, makes the first 32CA reaction irreversible; (iii) the attack of NI **1** on pyrazoline **6** is highly regioselective as **TS12r** is 3.1 kcal·mol^−1^ higher in Gibbs free energy than **TS12**; (iv) the first 32CA reaction via **TS11** is the rate-determining reaction of this domino process. The irreversible character of formation of pyrazoline **6**, together with the closer activation Gibbs free energy of **TS12** to that of **TS11**, allows the characterisation of pyrazoline **6** by ^1^H NMR spectra of the reaction medium [40]; (v) although **TS22** is 2.7 kcal·mol^−1^ below **TS12**, the corresponding reaction path is not operative since pyrazoline **14** is not formed along the first 32CA reaction; (vi) the strong exergonic character of the conversion of pyrazoline **6** in spirobipyrazoline **5** via **TS12**, −61.4 kcal·mol^−1^, makes this second 32CA reaction also irreversible; and finally, (vii) formation of the final spirobipyrazoline **5** is both kinetically and thermodynamically favourable.

The geometries of the TSs involved in the two competitive reaction paths of the domino reaction of diphenyl NI **1** with allenoate **4**, yielding spirobipyrazoline **5**, are given in Figure 4. At the TSs involved in the attack of diphenyl NI **1** on the *α*,*β*-conjugated C4–C5 double bond of allenoate **4**, the C1–C5 and N3–C4 distances are 2.191 and 2.580 Å at **TS11** and 2.087 and 2.458 Å at **TS22**, respectively, while at the TSs involved in the attack of diphenyl NI **1** on the terminal C5–C6 double bond of pyrazoline **6**, the N3–C4 and C1–C6 distances are 2.176 and 2.458 Å at **TS12** and 2.181 and 2.473 Å at **TS21**, respectively. Some appealing conclusions can be drawn from these geometrical parameters: (i) within some small differences, the four TSs show similar geometries, (ii) in the four cases, the shorter distance corresponds to the C1–C5(C6) one, involving the C1 carbon of diphenyl NI **1**; in the four TSs this distance ranges from 2.087 to 2.191 Å; (iii) considering that the C–C and C–N bond formation begins in the narrow range of distances of 2.0–1.9 and 1.9–1.8 Å, respectively [66], the distances between these interacting centers at these TSs suggest asynchronous bond formation processes in which the formation of the C–C single bond is more advanced than the N–C one, but in any case, none of them has yet begun; and finally, (iv) at the TSs involving the *α*,*β*-conjugated C4–C5 double bond, i.e., **TS11** and **TS22**, the shorter C–C distance corresponds to the participation of the most electrophilic center of the ethylene derivative **4**, and the C1 carbon of diphenyl NI **1**, which corresponds to the carbenoid center of the simplest NI **7** [42] but not to the most nucleophilic center of diphenyl NI **1** (see Section 3.2).

The polar character of these competitive 32CA reactions was evaluated analysing the GEDT at the four TSs related to the formation of spirobipyrazoline **5**. Reactions with GEDT values of 0.0 e correspond to non-polar processes, while values higher than 0.2 e correspond to polar processes. The GEDT, which fluxes from the NI framework to the allenoate one is 0.13 e at **TS11**, 0.07 e at **TS21**, 0.05 e at **TS12** and 0.10 e at **TS22**. These data indicate that while the reaction paths involving the *α*,*β*-conjugated C4–C5 double bond of allenoate **4** via **TS11** and **TS22** have some polar character, those involving the C5–C6 double bond via **TS21** and **TS12** have even lower polar character. As expected, the most polar **TS11** is the most favourable one [73].

### 3.4. ELF Characterisation of the Single Bond Formation along the Most Favourable Reaction Path Associated with the 32CA Reaction of Diphenyl NI **1** with Allenoate **4**

In order to understand how the formation of the C1–C5 and N3–C4 single bonds take place along the 32CA reaction between diphenyl NI **1** and allenoate **4**, a BET [64] study was carried out along the corresponding IRC to find the structures associated either with the formation of these two new single bonds or with relevant events for them to happen. The electron populations of the most relevant valence basins of the selected structures **S1**–**S9** as well as of **TS11** are gathered in Table 2, while the positions of these structures along the IRC are shown in Figure 5. The attractor positions of the ELF valence basins of the selected structures involved in the C1–C5 and N3–C5 single bond are shown in Figure 6.

The topological analysis of the ELF of **S1**, d(C1–C5) = 3.431 Å and d(N3–C4) = 3.235 Å, which is the first point of the IRC, is very similar to that of the separated reagents (see Section 3.1). Thus, at **S1**, ELF evidences the allenic structure of the allenoate framework, with two C4–C5 and C5–C5′ double bonds integrating total populations of 3.69 and 3.72 e, respectively, while the NI framework keeps presenting a C1–N2 triple bond integrating 5.69 e, an N2–N3 single bond with 2.13 e and a non-bonding electron population of 3.50 e at the N3 nitrogen.

At **S2**, d(C1–C5) = 3.421 Å and d(N3–C4) = 3.206 Å, due to the depopulation of the C1–N2 bonding region to 5.42 e, the C1 carbon of the NI framework gathers a non-bonding population of 0.28 e (see the V(C1) monosynaptic basin at **S2** in Table 2). This relevant topological change demands an unappreciable energy cost of only 0.3 kcal·mol^−1^.

At **TS11**, d(C1–C5) = 2.191 Å and d(N3–C4) = 2.580 Å, together with the strong depopulation of the adjacent C1–N2 triple bond by 1.99 e, which could already be considered an overpopulated double bond, a non-bonding region integrating 1.72 e is observed at the N2 nitrogen. Likewise, the C1 non-bonding region has acquired 0.89 e from the joint depopulation of both the C1–N2 and bonding region and the C1–Ph bond, whose population has decreased by 0.32 e. At the allenoate framework, the C4–C5 bonding region has been depopulated by 0.20 e, which is almost equally redistributed into the two C4–CO and C4–Me bonding regions.

At **S3**, d(C1–C5) = 2.075 Å and d(N3–C4) = 2.540 Å, a non-bonding region integrating 0.32 e is created at the C5 carbon of the allenoate moiety (see the V(C5) monosynaptic basin at **S3** in Table 2), coming mainly from the depopulation of the C4–C5 bonding region, which, with a population of 2.84 e, could already be considered an overpopulated single bond. At **S4**, d(C1–C5) = 1.958 Å and d(N3–C4) = 2.499 Å, the C5 non-bonding region has reached 0.55 e, while the population of the C1 non-bonding region has increased to 1.28 e.

At **S5**, d(C1–C5) = 1.947 Å and d(N3–C4) = 2.495 Å, while the two non-bonding regions present at the C1 and C5 carbons disappear, a bonding C1–C5 region integrating 1.85 e is created (see the presence of the V(C1,C5) disynaptic basin at **S5** in Table 2 and Figure 6). This topological change indicates that formation of the new C1–C5 single bond takes place at 1.95 Å by merging the non-bonding electron densities of both C1 and C5 carbons in a 70:30 ratio.

At **S6**, d(C1–C5) = 1.543 Å and d(N3–C4) = 2.234 Å, the joint depopulation of the C4–C5 and C4–CO bonding regions to 2.27 and 3.45 e prompts the creation of a small non-bonding region at the C4 carbon with a population of 0.19 e (see the V(C4) monosynaptic basin at **S6** in Table 2). The depopulation of the C4–C5 bonding region also contributes, together with the depopulation of the C5–C5′ bonding region to 3.45 e, to the increase of the population of the new C1–C5 single bond to 2.14 e, which can already be considered a single bond.

At **S7**, d(C1–C5) = 1.480 Å and d(N3–C4) = 1.928 Å, an interesting topological change is observed. Together with the disappearance of the C4 non-bonding region, the non-bonding population associated with the N3 nitrogen markedly increases to 3.37 e (see the V(N3) monosynaptic basin at **S7** in Table 2). This change suggests a local electron density transfer from the C4 carbon to the N3 nitrogen favoured by a retro-donation process from the allenoate moiety towards the NI one. Note that the GEDT at **S7** is −0.12 e.

At **S8**, d(C1–C5) = 1.479 Å and d(N3–C4) = 1.918 Å, while the non-bonding region of the N3 nitrogen is depopulated by 1.04 e, the second N3–C4 single bond is created with a population of 1.06 e (see the V(N3,C4) disynaptic basin at **S7** in Table 2 and Figure 6). This topological change indicates that the N3–C4 single bond is entirely formed at 1.92 Å by donation of part of the non-bonding electron density of the N3 nitrogen to the C4 carbon.

Finally, at **S9**, d(C1–C5) = 1.457 Å and d(N3–C4) = 1.486 Å, which is the last point of the IRC, the most noteworthy feature is the low population of the N2–N3 bonding region, 1.55 e. The non-bonding population of 1.93 e associated with the N3 nitrogen may suggest that the N2–N3 bonding region is polarised towards the N2 nitrogen, which presents 2.77 e, while the C1–N2 bonding region presents 3.03 e. The C4–C5 single bond ends up with 2.04 e, while the two new C1–C5 and N3–C4 single bonds reach 2.28 e and 1.66 e.

Some appealing conclusions can be drawn from this topological analysis of the ELF: (i) from **S1** to **TS11**, the most notable change in ELF valence basin populations involves the C1–N2 bonding region of the NI framework. Thus, the high activation energy on going from **S1** to **TS11**, 14.8 kcal·mol^−1^, could, in principle, be mainly related to the depopulation of the C1–N2 triple bond of diphenyl NI **1**; (ii) formation of the first C1–C5 single bond takes place at a C–C distance of ca. 1.95 Å, by sharing the non-bonding electron densities of the two C1 and C5 carbons in a 70:30 ratio (see Figure 6); (iii) the non-bonding population associated with the C1 carbon appears at a very early stage of the reaction, from the depopulation of the C1–N2 bond, demanding an energy cost of only 0.3 kcal·mol^−1^; (iv) formation of the second N3–C4 single bond takes place at an N–C distance of ca. 1.92 Å, by donation of some non-bonding electron density of the N3 nitrogen to the C4 carbon (see Figure 6); (v) since the formation of the first C1–C5 bond, there is a retro-GEDT process from the allenoate moiety towards the NI one, which somehow plays a role in the formation of the second N3–C4 single bond; (vi) formation of the second N3–C4 single bond begins when the formation of the first C1–C5 single bond is almost completed by up to 98%. Consequently, this 32CA reaction takes place through a *two-stage one-step* mechanism [74]; and finally, (vii) the present ELF topological analysis suggests that the molecular mechanism of the 32CA reaction between diphenyl NI **1** and allenoate **4** should be considered on the borderline between *pmr-type* and *cb-type* reaction mechanisms [66], somewhat closer to the latter.

## 4. Conclusions

The domino reaction of diphenyl NI **1** with methyl 1-methyl-allenoate **4** yielding spirobipyrazoline **5**, experimentally reported by Gou et al. [40], has been studied within MEDT at the MPWB1K/6-311G(d) computational level in DCM.

Unlike the simplest NI **7** and dimethyl NI **13**, which have a carbenoid structure, topological analysis of the ELF of diphenyl NI **1** indicates that this TAC has a zwitterionic structure, not presenting any *pseudoradical* nor carbenoid center. Analysis of the CDFT reactivity indices indicates that diphenyl NI **1** is a strong nucleophile, while allenoate **4** is a moderate electrophile. Analysis of the nucleophilic Parr functions at diphenyl NI **1** and the electrophilic Parr functions at allenoate **4** indicates that the most favourable reaction path should correspond to the two-center interaction between the N3 nitrogen of diphenyl NI **1** and the C5 carbon of allenoate **4**, which is against the experimental outcomes.

The reaction of diphenyl NI **1** with allenoate **4** yielding spirobipyrazoline **5** is a domino process that comprises two consecutive 32CA reactions of two molecules of diphenyl NI **1** with the two C5–C4(6) double bonds of allenoate **4**. Due to the non-symmetry of the two reagents, four competitive reaction paths, two chemoselective and two regioselective ones, are feasible for the first 32CA reaction, while the second 32CA reaction has only two competitive regioisomeric reaction paths. Analysis of the relative Gibbs free energies indicates that the first 32CA reaction is completely chemo- and highly regioselective, giving pyrazoline **6**, and the second 32CA reaction is highly regioselective yielding the final spirobipyrazoline **5**, in complete agreement with the mechanistic proposal of Gou et al., based on the experimental isolation of spirobipyrazoline **5** as the only product [40]. The first 32CA reaction is the rate-determining reaction in this domino process. The irreversible character of the formation of pyrazoline **6**, together with the similar activation Gibbs free energies associated to **TS12** and **TS2**, make the experimental characterisation of pyrazoline **6** in the reaction medium possible [40].

Analysis of the geometries of the TSs involved in this domino reaction indicates that they are associated to asynchronous bond formation processes in which the shorter distance involves the C1 carbon of diphenyl NI **1**. The most favourable two-center interaction between the C1 carbon of diphenyl NI **1** and the C5 carbon allenoate **4** accounts for the chemo- and regioselectivity found in the first 32CA reactions.

Topological analysis of the ELF of some selected structures of the IRC associated with the 32CA reaction of diphenyl NI **1** and allenoate **4** shows the presence of a *pseudoradical* structure at the beginning of the reaction path, appearing with a very low energy cost of 0.3 kcal·mol^−1^ and with an initial population of 0.28 e, which reaches 1.28 e before the formation of the first C1–C5 single bond. Formation of the second N3–C4 single bond begins when the formation of the first C1–C5 single bond is almost completed by up to 98%. Consequently, this 32CA reaction takes place through a *two-stage one-step* mechanism. The present ELF topological analysis suggests that the molecular mechanism of the 32CA reaction between diphenyl NI **1** and allenoate **4** should be considered on the borderline between *pmr-type* and *cb-type* reaction mechanisms, somewhat closer to the latter.

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
