# Peer review of "A Molecular Electron Density Theory Study of the Synthesis of Spirobipyrazolines through the Domino Reaction of Nitrilimines with Allenoates"

_molecules, 2019, doi:10.3390/molecules24224159_

Round 1

Reviewer 1 Report

In this manuscript by Domingo and co-workers, a molecular electron density theory study of the synthesis of spirobipyrazolines are described.

The paper is technically well-executed and useful from theoretical chemistry points of view. The manuscript is well-written and demanding, however the referee suggests some minor modifications.

there are quite a lot of typing mistakes.

e.g.: lines 69, 71, 91, 123, 145, 151, 154-156, etc.: a bond is missing among the atoms, the whole manuscript should be carefully checked and improved.

lines 55, 69, 245, 261, 264, 266, 279, 306, 318, 330, 466, 469: “α,β-conjugated” should be written instead of “-conjugated” line 70: 1 should be written in superscript the values should be written together with the percent sign. e.g.: schemes 1 and 2: “78%” instead of “78%” Scheme 2: The word “yield” should be deleted. the reference list is not unified, e.g. see references 3 and 10.

In summary I recommend that the paper be accepted for publication in Molecules after suggested minor modifications.

Author Response

We thanks all suggestions made by the Referee 1, which have improved the quality of the manuscript.

Lines 69, 71, 91, 123, 145, 151, 154-156, etc.: a bond is missing among the atoms, the whole manuscript should be carefully checked and improved.

R. All suggested changes have been done in this revised version.

lines 55, 69, 245, 261, 264, 266, 279, 306, 318, 330, 466, 469: “α,β-conjugated” should be written instead of “-conjugated” line 70: 1 should be written in superscript the values should be written together with the percent sign. e.g.: schemes 1 and 2: “78%” instead of “78%” Scheme 2: The word “yield” should be deleted. the reference list is not unified, e.g. see references 3 and 10.

R. All suggested changes have been done. In addition, the references have been checked and unified.

Reviewer 2 Report

(1) In the caption of Scheme 2, the designation of the allenoate should be 4, not 2. (2) In Line 144, the designation of NI should be 7, not 8. (3) In Line 157, the desigantion of methyl 1-methyl-allenoate should be 4, not 1. Please check the manuscript again carefully. In the other places, the authors mentioned they used the MPWB1K DFT functional. However, in the Computational Method section, they mentioned another DFT functional, B3LYP. There are some typos. For example:(1) In the caption of Table 1, it should be MPWB1K, not MWB1K. (2) In line 146, it should be pseudodiradical, not peudodiradical. Please check the manuscript again carefully. There must be a space between the value and the unit. Please check the manuscript again carefully. In the main document, some special symbols were disappeared. In Line 339, the authors mentioned they selected structures S1 ~ S8. However, in Lines 388~393, they discussed the results of structure S9. 

Author Response

We thanks all suggestions made by this Referee and careful reading of the manuscript which have improved the quality of the manuscript. As suggested, the manuscript has been carefully revised.

(1) In the caption of Scheme 2, the designation of the allenoate should be 4, not

R. The corresponding change has been done.

(2) In Line 144, the designation of NI should be 7, not 8.

R. The corresponding change has been done.

(3) In Line 157, the desigantion of methyl 1-methyl-allenoate should be 4, not 1. Please check the manuscript again carefully. In the other places, the authors mentioned they used the MPWB1K DFT functional. However, in the Computational Method section, they mentioned another DFT functional, B3LYP. There are some typos. For example:(1) In the caption of Table 1, it should be MPWB1K, not MWB1K.

R. We thank these referee’s suggestions. The corresponding changes have been done, and the manuscript has been carefully revised.

(4) In line 146, it should be pseudodiradical, not peudodiradical. Please check the manuscript again carefully. There must be a space between the value and the unit. Please check the manuscript again carefully. In the main document, some special symbols were disappeared. In Line 339, the authors mentioned they selected structures S1 ~ S8. However, in Lines 388~393, they discussed the results of structure S9.

R. We newly thanks these referee’s suggestions. The suggested changes have been done.